# Prediction models for post-discharge mortality among under-five children with suspected sepsis in Uganda: A multicohort analysis

Matthew O. Wiens[1,2,3,4]*, Vuong Nguyen[1], Jeffrey N. Bone[3], Elias Kumbakumba[5], Stephen Businge[6], Abner Tagoola[7], Sheila Oyella Sherine[8], Emmanuel Byaruhanga[9], Edward Ssemwanga[10], Celestine Barigye[11], Jesca Nsungwa[12], Charles Olaro[12], J. Mark Ansermino[1,2,3], Niranjan Kissoon[3,13], Joel Singer[14], Charles P. Larson[15], Pascal M. Lavoie[3,13], Dustin Dunsmuir[1,3], Peter P. Moschovis[16], Stefanie Novakowski[1,2], Clare Komugisha[4], Mellon Tayebwa[4], Douglas Mwesigwa[4], Martina Knappett[1], Nicholas West[3], Nathan Kenya Mugisha[4], Jerome Kabakyenga[17,18]*

1 Institute for Global Health at BC Children's and Women's Hospital, Vancouver, Canada, 2 Department of Anesthesiology, Pharmacology & Therapeutics, University of British Columbia, Vancouver, Canada, 3 BC Children's Hospital Research Institute, Vancouver, Canada, 4 Walimu, Kampala, Uganda, 5 Department of Paediatrics and Child Health, Mbarara University of Science and Technology, Mbarara, Uganda, 6 Holy Innocents Children's Hospital, Mbarara, Uganda, 7 Jinja Regional Referral Hospital, Jinja City, Uganda, 8 Masaka Regional Referral Hospital, Masaka, Uganda, 9 Kawempe National Referral Hospital, Kampala, Uganda, 10 Villa Maria Hospital, Masaka, Uganda, 11 Mbarara Regional Referral Hospital, Mbarara, Uganda, 12 Ministry of Health for the Republic of Uganda, Kampala, Uganda, 13 Department of Pediatrics, University of British Columbia, Vancouver, Canada, 14 School of Population and Public Health, University of British Columbia, Vancouver, Canada, 15 School of Population and Global Health, McGill University, Montréal, Canada, 16 Division of Global Health, Massachusetts General Hospital, Boston, MA, United States of America, 17 Maternal Newborn & Child Health Institute, Mbarara University of Science and Technology, Mbarara, Uganda, 18 Faculty of Medicine, Department of Community Health, Mbarara University of Science and Technology, Mbarara, Uganda

* matthew.wiens@bcchr.ca (MOW); jkabakyenga@must.ac.ug (JK)

## Abstract

In many low-income countries, over five percent of hospitalized children die following hospital discharge. The lack of available tools to identify those at risk of post-discharge mortality has limited the ability to make progress towards improving outcomes. We aimed to develop algorithms designed to predict post-discharge mortality among children admitted with suspected sepsis. Four prospective cohort studies of children in two age groups (0–6 and 6–60 months) were conducted between 2012–2021 in six Ugandan hospitals. Prediction models were derived for six-months post-discharge mortality, based on candidate predictors collected at admission, each with a maximum of eight variables, and internally validated using 10-fold cross-validation. 8,810 children were enrolled: 470 (5.3%) died in hospital; 257 (7.7%) and 233 (4.8%) post-discharge deaths occurred in the 0-6-month and 6-60-month age groups, respectively. The primary models had an area under the receiver operating characteristic curve (AUROC) of 0.77 (95%CI 0.74–0.80) for 0-6-month-olds and 0.75 (95% CI 0.72–0.79) for 6-60-month-olds; mean AUROCs among the 10 cross-validation folds were 0.75 and 0.73, respectively. Calibration across risk strata was good: Brier scores were 0.07 and 0.04, respectively. The most important variables included anthropometry and

**Data Availability Statement:** Study materials including the study protocol, consent form, data collection tools, de-identified participant data, data

dictionary, and the analysis code are available on request to the corresponding author (Matthew O. Wiens, matthew.wiens@bcchr.ca) or to the Institute for Global Health at BC Children's and Women's Hospital (Jessica Trawin, jessica. trawin@cw.bc.ca) or through the published protocol (https://doi.org/10.5683/SP3/QRUMNQ) and dataset (https://doi.org/10.5683/SP3/ REPMSY). Owing to the sensitive nature of clinical data, access to the de-identified data is granted on a case-by-case basis and will require the signing of a data sharing agreement.

**Funding:** The study was supported by funds from Grand Challenges Canada (MW; grant #TTS-1809-1939, https://www.grandchallenges.ca/), Thrasher Research Fund (MW; grant #13878, https://www. thrasherresearch.org/), BC Children's Hospital Foundation (https://www.bcchf.ca/), and Mining4Life (https://mining4life.org/). The funders had no role in study design, data collection and analysis, decision to publish, or preparation of the manuscript.

**Competing interests:** The authors have declared that no competing interests exist.

oxygen saturation. Additional variables included: illness duration, jaundice-age interaction, and a bulging fontanelle among 0-6-month-olds; and prior admissions, coma score, temperature, age-respiratory rate interaction, and HIV status among 6-60-month-olds. Simple prediction models at admission with suspected sepsis can identify children at risk of post-discharge mortality. Further external validation is recommended for different contexts. Models can be digitally integrated into existing processes to improve peri-discharge care as children transition from the hospital to the community.

## Introduction

Morbidity and mortality secondary to sepsis disproportionately affect children in low- and middle-income countries, where >85% of global cases and deaths occur [1]. Lower income regions are plagued by poorly resilient health systems, widespread socio-economic deprivation, and unique vulnerabilities, including malnutrition. Reducing the overall sepsis burden requires a multi-pronged strategy that addresses three periods along the care continuum–pre-facility, facility and post-facility [2]. Of these, post-facility issues have been largely neglected in research, policy, and practice [3].

Robust epidemiological data for pediatric post-discharge mortality in the context of sepsis and severe infection have been limited [4]. Growing evidence points to a significant burden of post-discharge mortality, which accounts for as many deaths as the acute hospital phase of illness [5,6]. While comorbid conditions such as malnutrition and anemia have been linked to risk, other factors such as illness severity (at admission and discharge), prior hospitalizations, and underlying social vulnerability, are also independently associated with poor post-discharge outcomes [7]. However, we lack simple data-driven methods to identify those at highest risk of mortality.

Current epidemiological evidence has demonstrated critical gaps in care following discharge [8]. Most post-discharge deaths occur at home, rather than during a subsequent readmission, indicating poor health utilization among the most vulnerable. Effective healthcare utilization is often hampered by poverty, community and family social dynamics, and poorly linked and unresponsive health facilities [9–11]. Providing quality care during and after discharge is a significant challenge in many facilities, in part due to severely strained human and material resources.

Effective solutions to improving the transition of care from hospital to home within poorly resourced health systems must be child-centred and focused on identifying the most vulnerable children [12]. In this study, we aim to update the development and validation of clinical prediction models that identify children, admitted with suspected sepsis, who are at risk of post-discharge mortality [13].

## Materials and methods

### Study design and approvals

Four independently funded, prospective observational cohort studies were conducted with a primary objective of generating model-building data: two among children under six months and two among children 6–60 months of age. These studies were approved by the Mbarara University of Science and Technology Research Ethics Committee (No. 05/11-11, 10-Nov-2011; and No. 15/10-16, 27-Jan-2017) and the University of British Columbia–Children's and

Women's Health Centre of BC Research Ethics Board (H10-01927, 01-Dec-2011; and H16-02679, 09-May-2017). Written informed consent was obtained from the parent or legal guardian of all study participants. This manuscript adheres to the Transparent Reporting of a multivariable prediction model for Individual Prognosis Or Diagnosis (TRIPOD) statement [14].

## Study setting and population

Subjects were enrolled from six hospitals in Uganda (**S1 Text**). These facilities serve catchments of 30 districts with a population of approximately 8.2 million individuals, including approximately 1.4 million children under five years [15], in a mix of urban and rural areas, reflecting a representative sampling of the Ugandan pediatric population.

All study cohorts had identical eligibility criteria. Any child admitted with suspected sepsis was eligible. Suspected sepsis was defined as children admitted with a proven or suspected infection (as determined by the treating medical team). We previously demonstrated that 90% of children enrolled using these criteria meet the international pediatric sepsis consensus conference (IPSCC) definition [16]. The IPSCC defines sepsis as the presence of the systemic inflammatory response syndrome alongside a suspected or proven infection.

The first cohort (enrolment 13-Mar-2012 to 13-Jan-2014) was used previously to report a predictive model for post-discharge mortality in 6-60-month-olds [13]. The second and third cohorts were the primary enrolment for the present analysis, and were defined by age range: 0-6-month-olds (enrolment 11-Jan-2018 to 30-Mar-2020) and 6-60-month-olds (enrolment 13-Jul-2017 to 02-Jul-2019); these data have been previously reported [6]. The fourth cohort enrolled only 0-6-month-olds (enrolment 31-Mar-2020 to 05-Aug-2021) in order to understand how the early COVID-19 period impacted post-discharge outcomes. Protocols and procedures were largely overlapping, and the same research staff were involved in data collection during all four enrolment periods [17].

## Data collection

Data collection tools are available through the Smart Discharges Dataverse [17]. Data collection procedures were previously described (also **S1 Text**) [6,13]. Briefly, trained study nurses collected clinical, social, and demographic data from consented participants at hospital admission; largely overlapping between the two age groups, some variables were specific to 0-6-month-olds. These variables were our candidate predictors and were selected based on clinical and contextual knowledge of possible factors relating to post-discharge mortality, using a modified Delphi process to identify promising variables in each age group [18,19].

Study nurses recorded discharge diagnosis and status (died, discharged, discharged against medical advice, referred). A field officer contacted enrolled children by phone two and four months after discharge, with an in-person visit at six months to determine mortality status and, if applicable, date of death. All data were collected using encrypted study tablets and uploaded to a Research Electronic Data Capture (REDCap) database hosted at the BC Children's Hospital Research Institute (Vancouver, Canada) [20,21].

## Model development

**Outcome definition and ascertainment.** The primary outcome of the prediction model was post-discharge mortality within six months of discharge, analyzed as a binary outcome. While data were available to build a time-to-event prediction model, time of death was considered irrelevant for modelling mortality. Complete six-month follow-up data for vital status was available for 98% of our cohort.

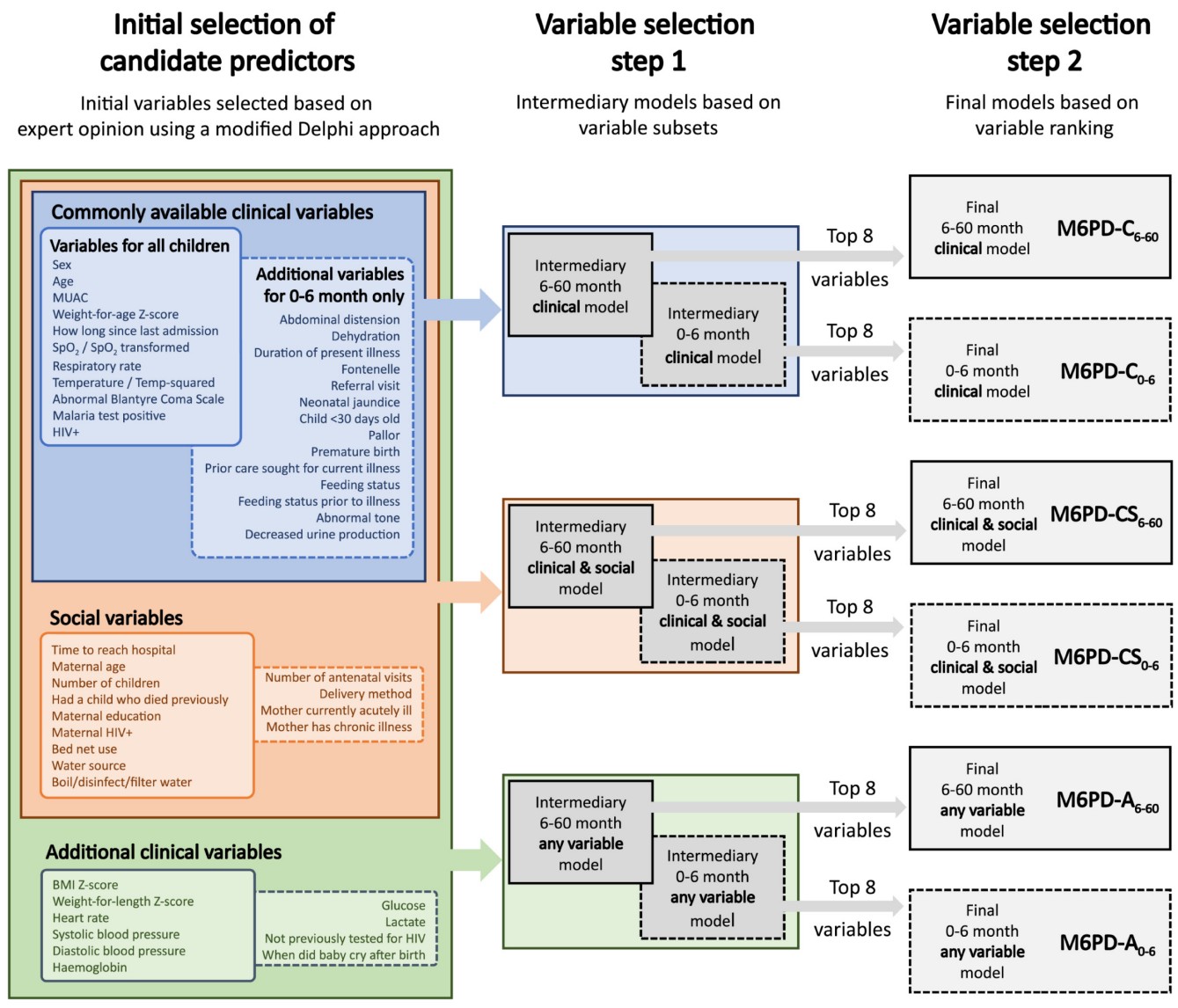

**Fig 1. Variable selection for model development.**

Prediction model performance was evaluated primarily using area under the receiver operating characteristic curve (AUROC). We also reported the precision-recall curve and area under the precision-recall curve (PR-AUC), which are more appropriate for imbalanced datasets [22].

**Variable selection.** Recognising the challenges of implementing large prediction models in resource-constrained settings, we determined *a priori* to develop three models for each age group and restricted each model to eight variables drawing from a different pool of available predictors: one model focused solely on commonly-available *clinical* variables; one model focused on commonly-available *clinical and social* variables; and one model used *any* candidate predictor variable (**Fig 1**). This approach aimed to reduce the impact of missingness in an implementation scenario. A feature of our modelling approach (elastic net regression) was that

final model size could not be pre-specified, often resulting in large models. Therefore, we conducted two rounds of variable selection.

To prioritize parsimony, the first variable selection round reduced the list of possible predictors to two subsets: one including only the most relevant *clinical* variables; and a second including only the most relevant *clinical and social* variables. Variables included in these subsets were determined *a priori*, based on clinical significance and ease of measurement in low-resource settings. These subsets were used to derive **intermediary models** that were either clinically-focused or clinically- and socially-focused; the full candidate predictor list for each age group was also used to derive **intermediary models** that used any available variable (**Tables B and C in S1 Text**).

The second variable selection round involved ranking the importance of variables from each intermediary model, which was calculated as the weighted sums of the absolute regression coefficients [23]. The top eight unique variables (e.g., temperature and its quadratic term were considered a single unique variable) were selected based on average ranking from 10-fold cross-validation of the intermediary models. If an interaction term was ranked in the top eight variables, both interaction terms were included. This second variable selection round produced a family of **final models** to predict mortality within six months post-discharge (M6PD) that used only the eight top-ranked variables in each age group: models using only *clinical* variables, denoted by $M6PD\text{-}C_{0\text{-}6}$ for 0-6-month-olds and $M6PD\text{-}C_{6\text{-}60}$ for 6-60-month-olds; models using *clinical and social* variables, denoted by $M6PD\text{-}CS_{0\text{-}6}$ and $M6PD\text{-}CS_{6\text{-}60}$; and models using *any* of available predictor variable, denoted by $M6PD\text{-}A_{0\text{-}6}$ and $M6PD\text{-}A_{6\text{-}60}$ (**Fig 1**).

## Statistical analysis

The primary study sample size was determined to accomplish three aims: to explore the epidemiology of post-discharge mortality, as previously reported [6]; to develop prediction models; and as a control period for a later interventional phase. The estimated sample size was determined as 2,117 and 1,551 for the 0-6-month and 6-60-month cohorts, respectively (**S2 Text**). All analyses were conducted using R version 4.2.2 (R Foundation for Statistical Computing, Vienna, Austria) [24], reported in detail in **S2 Text**.

## Results

### Study population

During the four enrolment periods, a total of 22,166 consecutively admitted children were screened and 8,810 enrolled (**Fig 2**). Among 0-6-month-olds (n = 3,665), 3,424 (93.4%) survived to discharge. Complete 6-month outcomes were available for 3,349 (97.8%) of these children, forming the full dataset for model development in this age group. Among 6-60-month-olds (n = 5,145), 4,916 (95.5%) survived to discharge. Complete 6-month outcomes were available for 4,830 (98.2%) of these children, forming the full dataset for model development in this age group.

Mortality within 6 months of discharge occurred in 257 (7.7%) 0-6-month-olds, with median (interquartile range [IQR]) time to death of 31 (9–80) days, and in 233 (4.8%) 6-60-month-olds, with time to death of 36 (11–105) days (**Fig A in S1 Text**). Missing data were minimal (**Table 1**).

These cohorts' clinical and demographic details have been previously described (**Table 1**) [6,13]. The mean ±standard deviation [SD] age was 2.1 ±1.8 months with 1,884 (56.3%) male in the 0-6-month group, and 21.7 ±13.7 months with 2,670 (55.3%) male in the 6-60-month group. Poor growth/malnutrition was common, with 463 (13.8%) 0-6-month-olds and 668

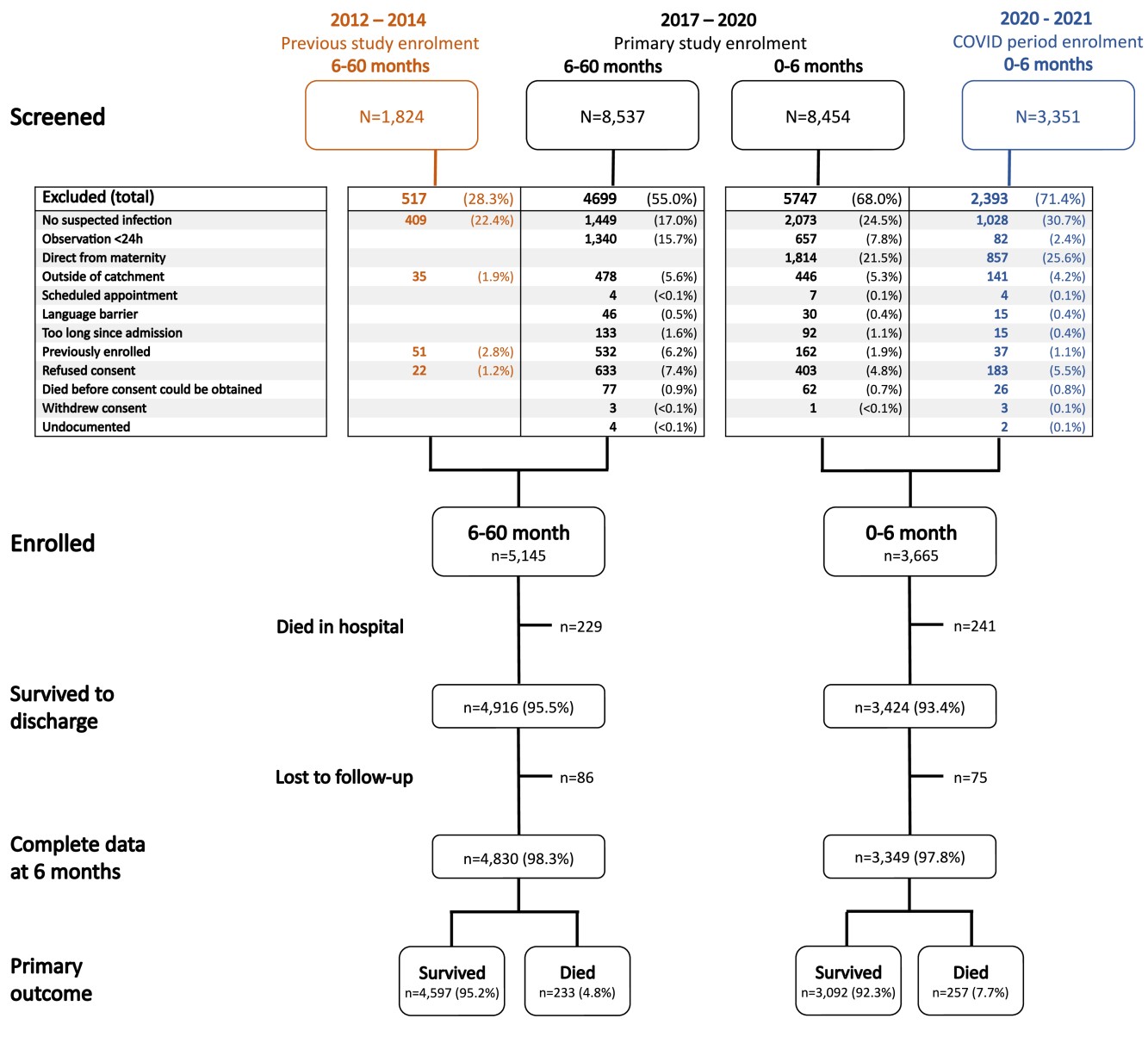

**Fig 2. Study enrolment flow diagram.**

(13.8%) 6-60-month-olds classified as severely underweight (weight-for-age z-score <-3) and similar weight-for-age z-score distributions in both age groups. Discharge diagnoses recorded by the clinical team could be overlapping in the case of multiple diagnoses (**Table D in S1 Text**). Most predictor variables considered were associated with post-discharge mortality (**Table 1**).

## Prediction models

The intermediary variable models were large (coefficients, performance metrics, and variable importance reported in **S3**–**S5 Texts**). The models derived using all candidate predictors (intermediary *any* variable model) included 41 unique variables in the 0-6-month model and 19 unique variables in the 6-60-month model. Applied to the entire dataset for each age group,

**Table 1. Demographics and univariable odds ratios for the risk of post-discharge infant mortality.**

| Variable | 0–6 month (n = 3349) | | | | 6–60 month (n = 4830) | | | |
|---|---|---|---|---|---|---|---|---|
| | n (%)/Mean (SD) | n Missing (%) | OR (95% CI) | P-value | n (%)/Mean (SD) | n Missing (%) | OR (95% CI) | P-value |
| **A) Demographics** | | | | | | | | |
| **Sex, male** | 1884 (56.3%) | 0 (0%) | 1.18 (0.91, 1.53) | 0.218 | 2670 (55.3%) | 0 (0%) | 0.9 (0.69, 1.17) | 0.433 |
| **Age, months** | 2.1 (1.8) | 0 (0%) | 1.05 (0.98, 1.12) | 0.188 | 21.7 (13.7) | 1 (0.02%) | 1 (0.99, 1.01) | 0.471 |
| **B) Admission Anthropometry** | | | | | | | | |
| **BMI Z-scores** | -1 (2.2) | 5 (0.15%) | 0.78 (0.74, 0.82) | <**0.001** | -1 (9.8) | 32 (0.66%) | 0.86 (0.81, 0.91) | <**0.001** |
| *< -3* | 565 (16.9%) | | 4.31 (3.24, 5.75) | <**0.001** | 775 (16%) | | 2.31 (1.68, 3.16) | <**0.001** |
| *-3 to -2* | 399 (11.9%) | | 2.36 (1.61, 3.40) | <**0.001** | 684 (14.2%) | | 1.84 (1.28, 2.60) | **0.001** |
| *> -2* | 2380 (71.1%) | | ref. | <**0.001** | 3339 (69.1%) | | ref. | <**0.001** |
| MUAC, mm [a] | 113.7 (17.7) | 3 (0.09%) | 0.96 (0.96, 0.97) | <**0.001** | 139.2 (16.1) | 18 (0.37%) | 0.96 (0.95, 0.97) | <**0.001** |
| *<110 / <115* | 1304 (38.9%) | | 3.81 (2.7, 5.51) | <**0.001** | 321 (6.6%) | | 6.66 (4.76, 9.25) | <**0.001** |
| *110–120 / 115–125* | 942 (28.1%) | | 1.58 (1.04, 2.42) | **0.033** | 514 (10.6%) | | 2.77 (1.92, 3.92) | <**0.001** |
| *>120 / >125* | 1100 (32.8%) | | ref. | <**0.001** | 3977 (82.3%) | | ref. | <**0.001** |
| **Weight for age Z-scores** | -1.1 (2) | 2 (0.06%) | 0.71 (0.67, 0.75) | <**0.001** | -1.3 (1.7) | 12 (0.25%) | 0.71 (0.66, 0.76) | <**0.001** |
| *< -3* | 463 (13.8%) | | 6.15 (4.58, 8.26) | <**0.001** | 668 (13.8%) | | 4.58 (3.40, 6.17) | <**0.001** |
| *-3 to -2* | 356 (10.6%) | | 3.61 (2.51, 5.14) | <**0.001** | 723 (15%) | | 1.77 (1.20, 2.55) | **0.003** |
| *> -2* | 2528 (75.5%) | | ref. | <**0.001** | 3427 (71%) | | ref. | <**0.001** |
| **Weight for length Z-scores** | -1 (2.6) | 5 (0.15%) | 0.87 (0.84, 0.91) | <**0.001** | -1.2 (2) | 30 (0.62%) | 0.83 (0.78, 0.89) | <**0.001** |
| *< -3* | 627 (18.7%) | | 2.52 (1.88, 3.35) | <**0.001** | 725 (15%) | | 2.52 (1.83, 3.45) | <**0.001** |
| *-3 to -2* | 365 (10.9%) | | 1.73 (1.16, 2.52) | **0.006** | 718 (14.9%) | | 1.86 (1.30, 2.61) | <**0.001** |
| *> -2* | 2352 (70.2%) | | ref. | <**0.001** | 3357 (69.5%) | | ref. | <**0.001** |
| **C) Admission Clinical Assessment** | | | | | | | | |
| **How long ago since last admission** | | 20 (0.6%) | | | | 20 (0.41%) | | |
| *Never* | 2848 (85%) | | ref. | <**0.001** | 2647 (54.8%) | | ref. | <**0.001** |
| *< 7days* | 122 (3.6%) | | 2.04 (1.12, 3.47) | **0.013** | 191 (4%) | | 2.37 (1.34, 3.94) | **0.002** |
| *7 days to <1 month* | 180 (5.4%) | | 2.68 (1.71, 4.06) | <**0.001** | 400 (8.3%) | | 2.39 (1.60, 3.52) | <**0.001** |
| *1 month to <1 year* | 179 (5.3%) | | 2.47 (1.56, 3.79) | <**0.001** | 1175 (24.3%) | | 1.42 (1.03, 1.94) | **0.031** |
| *≥1 year* | 0 (0%) | | | | 397 (8.2%) | | 0.50 (0.22, 0.97) | 0.06 |
| SpO$_2$ | 93.8 (6.8) | 9 (0.27%) | 0.96 (0.95, 0.98) | <**0.001** | 94.2 (6.5) | 22 (0.46%) | 0.95 (0.94, 0.97) | <**0.001** |
| *< 90%* | 598 (17.9%) | | 1.78 (1.31, 2.41) | <**0.001** | 774 (16%) | | 2.07 (1.49, 2.84) | <**0.001** |

(*Continued*)

**Table 1.** (Continued)

| | 0–6 month (n = 3349) | | | | 6–60 month (n = 4830) | | | |
|---|---|---|---|---|---|---|---|---|
| *90% to 95%* | 891 (26.6%) | | 0.87 (0.62, 1.20) | 0.406 | 1236 (25.6%) | | 1.15 (0.82, 1.58) | 0.404 |
| *> 95%* | 1851 (55.3%) | | ref. | <**0.001** | 2798 (57.9%) | | ref. | <**0.001** |
| **Heart rate, beats per minute** | 149.2 (23.6) | 3 (0.09%) | 1.00 (0.99, 1.00) | 0.276 | 144.8 (25.5) | 3 (0.06%) | 1.00 (0.99, 1.00) | 0.599 |
| **Respiratory rate, breaths per minute** | 57.4 (17) | 5 (0.15%) | 1 (0.99, 1.01) | 0.875 | 48.1 (15.7) | 7 (0.14%) | 1.01 (1.00, 1.02) | **0.003** |
| **Systolic blood pressure, mmHg** | 85.1 (16.5) | 10 (0.3%) | 0.99 (0.99, 1.00) | 0.08 | 95.2 (13.4) | 8 (0.17%) | 0.99 (0.98, 1.00) | **0.028** |
| **Diastolic blood pressure, mmHg** | 46.3 (12.8) | 10 (0.3%) | 0.99 (0.98, 1.00) | 0.213 | 54.4 (11.6) | 8 (0.17%) | 0.99 (0.98, 1.00) | 0.079 |
| **Temperature, ˚C** | 37.4 (0.9) | 1 (0.03%) | 0.90 (0.78, 1.04) | 0.167 | 37.7 (1.2) | 3 (0.06%) | 0.81 (0.72, 0.91) | <**0.001** |
| *< 36.5* | 386 (11.5%) | | 0.96 (0.62, 1.43) | 0.835 | 505 (10.5%) | | 1.28 (0.84, 1.89) | 0.234 |
| *36.5 to 37.5* | 1699 (50.7%) | | ref. | 0.581 | 1868 (38.7%) | | ref. | **0.014** |
| *37.6 to 39* | 1072 (32%) | | 1.01 (0.76, 1.34) | 0.923 | 1638 (33.9%) | | 0.82 (0.6, 1.12) | 0.222 |
| *> 39* | 191 (5.7%) | | 0.65 (0.32, 1.20) | 0.202 | 816 (16.9%) | | 0.58 (0.37, 0.89) | **0.016** |
| **Abnormal BCS score** | 285 (8.5%) | 0 (0%) | 2.37 (1.64, 3.34) | <**0.001** | 408 (8.4%) | 0 (0%) | 1.93 (1.30, 2.78) | **0.001** |
| **Malaria test positive** | 324 (9.7%) | 1 (0.03%) | 0.56 (0.31, 0.92) | **0.032** | 1480 (30.6%) | 11 (0.23%) | 0.76 (0.56, 1.02) | 0.075 |
| **HIV+** | 119 (3.6%) | 2 (0.06%) | 1.37 (0.70, 2.42) | 0.317 | 144 (3%) | 22 (0.46%) | 3.81 (2.31, 6.00) | <**0.001** |
| **Haemoglobin, g/dL** | 13 (3.3) | 4 (0.12%) | 0.96 (0.92, 1.00) | **0.036** | 10.4 (3.2) | 608 (12.59%) [b] | 0.88 (0.85, 0.92) | <**0.001** |
| *No anaemia* | 2435 (72.7%) | | ref. | **0.003** | 1983 (41.1%) | | ref. | <**0.001** |
| *Mild anaemia* | 788 (23.5%) | | 1.29 (0.96, 1.72) | 0.091 | 1535 (31.8%) | | 1.59 (1.15, 2.21) | **0.006** |
| *Severe anaemia* | 122 (3.6%) | | 2.47 (1.44, 4.04) | **0.001** | 704 (14.6%) | | 2.67 (1.87, 3.82) | <**0.001** |
| **D) Maternal and Social Characteristics** | | | | | | | | |
| **Time it took to reach hospital** | | 0 (0%) | | | | 1 (0.02%) | | |
| *<30 minutes* | 806 (24.1%) | | ref. | <**0.001** | 1015 (21%) | | ref. | <**0.001** |
| *30 minutes to <1 hour* | 1224 (36.5%) | | 1.15 (0.77, 1.75) | 0.498 | 1519 (31.4%) | | 1.67 (1.04, 2.75) | **0.037** |
| *≥1 hour* | 1319 (39.4%) | | 2.65 (1.86, 3.88) | <**0.001** | 2295 (47.5%) | | 2.89 (1.90, 4.58) | <**0.001** |
| **Maternal age, years** | 26.3 (5.7) | 50 (1.49%) | 1.00 (0.98, 1.02) | 0.871 | 27.9 (6.4) | 167 (3.46%) | 1.00 (0.98, 1.02) | 0.899 |
| **Number of children** | 2.8 (1.8) | 1 (0.03%) | 1.01 (0.94, 1.08) | 0.87 | 3.2 (2.1) | 3 (0.06%) | 1.04 (0.98, 1.10) | 0.228 |
| **Had a child who died previously** | 577 (17.2%) | 1 (0.03%) | 1.21 (0.87, 1.65) | 0.249 | 1066 (22.1%) | 3 (0.06%) | 1.27 (0.93, 1.70) | 0.123 |
| **Maternal education** | | 16 (0.48%) | | | | 49 (1.01%) | | |
| *No school* | 105 (3.1%) | | ref. | **0.001** | 334 (6.9%) | | ref. | <**0.001** |
| *≤P3* | 207 (6.2%) | | 1.16 (0.57, 2.48) | 0.684 | 345 (7.1%) | | 1.30 (0.70, 2.43) | 0.411 |
| *P4-P7* | 1327 (39.6%) | | 0.71 (0.39, 1.41) | 0.297 | 2088 (43.2%) | | 0.98 (0.61, 1.65) | 0.922 |

(*Continued*)

**Table 1.** (Continued)

| | 0–6 month (n = 3349) | | | | 6–60 month (n = 4830) | | | |
|---|---|---|---|---|---|---|---|---|
| S1-S6 | 1175 (35.1%) | | 0.58 (0.32, 1.16) | 0.098 | 1517 (31.4%) | | 0.61 (0.36, 1.07) | 0.073 |
| Post-Secondary | 519 (15.5%) | | 0.36 (0.18, 0.77) | **0.006** | 497 (10.3%) | | 0.41 (0.19, 0.85) | **0.018** |
| **Maternal HIV** | | 1 (0.03%) | | | | 6 (0.12%) | | |
| No | 3052 (91.1%) | | ref. | 0.125 | 3915 (81.1%) | | ref. | **0.012** |
| Yes | 246 (7.3%) | | 1.48 (0.95, 2.24) | 0.071 | 432 (8.9%) | | 1.63 (1.07, 2.40) | **0.016** |
| Unknown | 50 (1.5%) | | 1.71 (0.65, 3.77) | 0.222 | 477 (9.9%) | | 1.57 (1.05, 2.29) | **0.023** |
| **Bed net use** | | 1 (0.03%) | | | | 3 (0.06%) | | |
| Never | 2809 (83.9%) | | 0.85 (0.44, 1.57) | 0.611 | 3630 (75.2%) | | 1.05 (0.64, 1.73) | 0.841 |
| Sometimes | 337 (10.1%) | | ref. | 0.544 | 631 (13.1%) | | ref. | 0.443 |
| Always | 202 (6%) | | 0.80 (0.55, 1.20) | 0.262 | 566 (11.7%) | | 0.85 (0.59, 1.26) | 0.389 |
| **Water source** | | 0 (0%) | | | | 3 (0.06%) | | |
| Bore hole | 655 (19.6%) | | 1.72 (1.22, 2.4) | **0.002** | 1042 (21.6%) | | 2.61 (1.84, 3.72) | **<0.001** |
| Fast running water | 21 (0.6%) | | 0.84 (0.05, 4.08) | 0.862 | 515 (10.7%) | | 1.76 (1.08, 2.80) | **0.019** |
| Municipal water | 1630 (48.7%) | | ref. | **<0.001** | 1981 (41%) | | ref. | **<0.001** |
| Open source | 541 (16.2%) | | 2.12 (1.51, 2.98) | **<0.001** | 558 (11.6%) | | 1.88 (1.19, 2.93) | **0.006** |
| Protected spring | 392 (11.7%) | | 1.49 (0.97, 2.23) | 0.063 | 590 (12.2%) | | 2.03 (1.30, 3.11) | **0.001** |
| Slow running water | 110 (3.3%) | | 1.67 (0.80, 3.16) | 0.14 | 141 (2.9%) | | 1.99 (0.86, 4.03) | 0.075 |
| **Boil/disinfect/filter water** | 2526 (75.4%) | 0 (0%) | 0.84 (0.64, 1.13) | 0.237 | 3402 (70.4%) | 2 (0.04%) | 0.51 (0.39, 0.67) | **<0.001** |
| **E) Discharge Characteristics** | | | | | | | | |
| **Length of stay, days** | 5.6 (4.4) | 0 (0%) | | | 5.1 (8.2) | 0 (0%) | | |
| **Discharge status** | | 2 (0.06%) | | | | 0 (0%) | | |
| Referred to higher level of care | 164 (4.9%) | | | | 101 (2.1%) | | | |
| Routine discharge | 2810 (83.9%) | | | | 4143 (85.8%) | | | |
| Unplanned discharge | 373 (11.1%) | | | | 586 (12.1%) | | | |
| **F) Variables collected only for 0-6-month** | | | | | | | | |
| **Abdominal distension** | 217 (6.5%) | 2 (0.06%) | 1.79 (1.15, 2.70) | **0.007** | | | | |
| **Antenatal visits** | 4.9 (1) | 48 (1.43%) | 0.89 (0.78, 1.01) | 0.066 | | | | |
| **Dehydration, WHO categories** | | 11 (0.33%) | | | | | | |
| No dehydration | 2844 (84.9%) | | ref. | **<0.001** | | | | |
| Some dehydration | 399 (11.9%) | | 1.64 (1.15, 2.30) | **0.005** | | | | |
| Severe dehydration | 95 (2.8%) | | 3.40 (1.96, 5.62) | **<0.001** | | | | |
| **Delivery method, caesarean** | 497 (14.8%) | 4 (0.12%) | 0.74 (0.49, 1.08) | 0.136 | | | | |
| **Duration of present illness** | | 4 (0.12%) | | | | | | |
| <48 hours | 957 (28.6%) | | ref. | **<0.001** | | | | |

(Continued)

**Table 1.** (Continued)

| | 0–6 month (n = 3349) | | | 6–60 month (n = 4830) | | | |
|---|---|---|---|---|---|---|---|
| *48 hours to 7 days* | 60 (1.8%) | | 1.46 (1.05, 2.06) | **0.026** | | | | |
| *8 days to 1 month* | 1985 (59.3%) | | 3.16 (2.09, 4.80) | **<0.001** | | | | |
| *>1 month* | 343 (10.2%) | | 5.13 (2.52, 9.87) | **<0.001** | | | | |
| **Fontanelle** | 132 (3.9%) | 6 (0.18%) | 2.4 (1.44, 3.81) | **<0.001** | | | | |
| **Glucose, mmol/L** | 5.7 (2.5) | 2 (0.06%) | 1.03 (0.98, 1.08) | 0.188 | | | | |
| **Not previously tested for HIV** | 2968 (88.6%) | 0 (0%) | 0.93 (0.64, 1.40) | 0.719 | | | | |
| **Referral visit** | 1056 (31.5%) | 1 (0.03%) | 1.70 (1.31, 2.20) | **<0.001** | | | | |
| **Neonatal jaundice** | 261 (7.8%) | 34 (1.02%) | 1.31 (0.83, 1.99) | 0.219 | | | | |
| **Lactate level, mmol/L** | 2.5 (1.6) | 9 (0.27%) | 1.10 (1.03, 1.18) | **0.003** | | | | |
| **Mother currently acutely ill** | 132 (3.9%) | 13 (0.39%) | 0.56 (0.22, 1.18) | 0.171 | | | | |
| **Mother has chronic illness** | 251 (7.5%) | 19 (0.57%) | 1.29 (0.81, 1.97) | 0.256 | | | | |
| **Child less than 30 days old** | 1353 (40.4%) | 2 (0.06%) | 0.68 (0.52, 0.89) | **0.006** | | | | |
| **Pallor** | 307 (9.2%) | 2 (0.06%) | 2.15 (1.50, 3.03) | **<0.001** | | | | |
| **Premature birth** | 210 (6.3%) | 6 (0.18%) | 2.05 (1.33, 3.06) | **0.001** | | | | |
| **Prior care sought for current illness** | 1995 (59.6%) | 0 (0%) | 1.82 (1.38, 2.42) | **<0.001** | | | | |
| **Sucking well when breastfeeding, or feeding well if not breastfed** | 1956 (58.4%) | 8 (0.24%) | 0.47 (0.36, 0.61) | **<0.001** | | | | |
| **Sucking well when breastfeeding, or feeding well if not breastfed, prior to illness** | 2589 (77.3%) | 392 (11.7%) | 0.59 (0.42, 0.85) | **0.004** | | | | |
| **When did the baby cry after birth** | | 97 (2.9%) | | | | | | |
| *Immediately* | 2805 (83.8%) | | ref. | **0.041** | | | | |
| *<5 minutes* | 138 (4.1%) | | 1.83 (1.04, 3.02) | **0.025** | | | | |
| *5 to 10 minutes* | 141 (4.2%) | | 1.32 (0.70, 2.30) | 0.351 | | | | |
| *11 to 30 minutes* | 68 (2%) | | 1.26 (0.48, 2.72) | 0.594 | | | | |
| *>30 minutes* | 100 (3%) | | 2.12 (1.14, 3.68) | **0.011** | | | | |
| **Abnormal tone** | 285 (8.5%) | 2 (0.06%) | 3.11 (2.21, 4.31) | **<0.001** | | | | |
| **Decreased urine production** | 677 (20.2%) | 99 (2.96%) | 1.91 (1.43, 2.52) | **<0.001** | | | | |

For non-binary categorical variables, the p-value for the reference group (labelled ref.) indicates the global p-value. Odds ratios and p-values were not calculated for discharge variables.

[a] MUAC thresholds given for 0-6-month / 6-60-month cohorts.

[b] High prevalence of missing data for hemoglobin due to faulty capillary tubes during data collection.

Abbreviations: BCS = Blantyre coma scale; BMI = body mass index; HIV+ = human immunodeficiency virus positive; MUAC = mid-upper arm circumference; OR = odds ratio; SpO$_2$ = oxygen saturation; WHO = World Health Organization.

the AUROC was 0.81 (95%CI 0.79 to 0.84) for the 0-6-month model and 0.79 (95%CI 0.77 to 0.82) for the 6-60-month model, with average AUROCs of 0.77 (range 0.69–0.87) and 0.76 (range 0.71–0.81) across the 10 cross-validations, respectively; the PR-AUC was 0.27 for the 0-6-month model and 0.18 for the 6-60-month model, with average PR-AUCs of 0.22 (range 0.13–0.31) and 0.16 (range 0.11–0.21) across the 10 cross-validations, respectively. Calibration was good at low predicted probabilities, with a Brier scores of 0.07 (range 0.06–0.07) for the 0-6-month model and 0.04 (range 0.04–0.05) for the 6-60-month model. Calibration decreased at higher predicted probabilities, although there were almost no individuals with probabilities >40%. In both age groups, mid-upper arm circumference (MUAC) was identified as the variable with the highest importance.

The final models are summarized in **Table 2**, and detailed in **S6–S8 Texts**, including all model terms, their coefficients, and plots outlining the relative importance of coefficients in each model.

## Final 0-6-month models

The M6PD-$C_{0-6}$ model, using only simple *clinical* variables, included weight-for-age z-score (mean rank $[r_m]$ = 1.4, selection frequency $[s_f]$ = 10), MUAC ($r_m$ = 1.6, $s_f$ = 10), feeding status ($r_m$ = 3.4, $s_f$ = 10), SpO$_2$ ($r_m$ = 5.8, $s_f$ = 9), duration of illness ($r_m$ = 6.2, $s_f$ = 9), age × jaundice ($r_m$ = 7.8, $s_f$ = 7), and bulging fontanelle ($r_m$ = 8.3, $s_f$ = 8) (**Table A in S3 Text**). The AUROC was 0.77 (95%CI 0.74 to 0.80) and PR-AUC was 0.23 when applied to the entire 0-6-month dataset (**Fig 3**), while the average AUROC and PR-AUC across the internal 10 cross-validations were 0.75 (range 0.63–0.85) and 0.23 (range 0.11–0.33), respectively (**Table 2A and S6 Text**). Setting the sensitivity to 80%, the corresponding probability threshold was 0.058; at this threshold, positive and negative predictive values were 14% and 97%, respectively. Calibration at low predicted probabilities was good, with a Brier score of 0.07 (**Fig 3 and Fig C in S6 Text**). Calibration at probabilities beyond 30–40% was poor, but sample sizes were very small in this range.

The M6PD-$CS_{0-6}$ model, using *social and clinical* variables, was nearly identical in performance to M6PD-$C_{0-6}$; the variables were largely overlapping with only fontanelle status replaced by travel time required to reach hospital (**Table 2A and S7 Text**). M6PD-$A_{0-6}$ that used *any* available variable, was identical to M6PD-$CS_{0-6}$ (**Table 2A and S8 Text**).

## Final 6-60-month models

The M6PD-$C_{6-60}$ model, using only *clinical* predictors, included nine variables (the 8[th] best-performing variable included an interaction with a new variable; **Table B in S3 Text**): MUAC ($r_m$ = 1, $s_f$ = 10), SpO$_2$ ($r_m$ = 2.7, $s_f$ = 10), weight-for-age z-score ($r_m$ = 2.8, $s_f$ = 10), time since prior admission ($r_m$ = 4.7, $s_f$ = 10), abnormal coma score ($r_m$ = 5.8, $s_f$ = 9), temperature ($r_m$ = 6.4, $s_f$ = 9), HIV status ($r_m$ = 6.5, $s_f$ = 9) and age × respiratory rate ($r_m$ = 9.1, $s_f$ = 2). The AUROC was 0.74 (95%CI 0.72 to 0.79) and PR-AUC was 0.17 when applied to the entire 6-60-month dataset (**Fig 4**), with an average AUROC of 0.73 (range 0.67–0.77) and average PR-AUC of 0.16 (range 0.10–0.19) across the 10 cross-validations (**Table 2B and S6 Text**). Setting sensitivity to 80%, the corresponding probability threshold was 0.036; at this threshold, positive and negative predictive values were 0.08 and 0.98, respectively. Calibration across risk strata was good with a Brier score of 0.04 (**Fig 4 and Fig D in S6 Text**).

The M6PD-$CS_{6-60}$ model, which used *clinical and social* variables, was almost identical to M6PD-$C_{6-60}$, with only home water source and water disinfection practices replacing coma score (**Table 2B and S7 Text**). M6PD-$A_{6-60}$ was similar to M6PD-$CS_{6-60}$, with water

**Table 2. Summary of performance and variables included in the set of final models with reduced number of variables using the probability threshold that gave a sensitivity of 0.8.**

| A) 0-6-month models | M6PD-C$_{0-6}$ | M6PD-CS$_{0-6}$ * | M6PD-A$_{0-6}$ * |
|---|---|---|---|
| **Average Cross-Validation Performance** | | | |
| Specificity | 0.60 | 0.61 | 0.61 |
| AUROC | 0.75 | 0.76 | 0.76 |
| PPV | 0.15 | 0.16 | 0.16 |
| NPV | 0.97 | 0.97 | 0.97 |
| PRAUC | 0.23 | 0.23 | 0.23 |
| Brier Score | 0.07 | 0.07 | 0.07 |
| **Full Dataset Performance** | | | |
| Specificity | 0.58 | 0.62 | 0.62 |
| AUROC | 0.77 | 0.77 | 0.77 |
| PPV | 0.14 | 0.15 | 0.15 |
| NPV | 0.97 | 0.97 | 0.97 |
| PRAUC | 0.23 | 0.22 | 0.22 |
| Brier Score | 0.07 | 0.07 | 0.07 |
| **Variables** | | | |
| Age, months | ✓ | ✓ | ✓ |
| Duration of present illness, categorical | ✓ | ✓ | ✓ |
| MUAC, mm | ✓ | ✓ | ✓ |
| Neonatal jaundice, binary | ✓ | ✓ | ✓ |
| Sucking well when breastfeeding, binary | ✓ | ✓ | ✓ |
| SpO$_2$, % | ✓ | ✓ | ✓ |
| Time to reach hospital, categorical | | ✓ | ✓ |
| Weight for age z-score | ✓ | ✓ | ✓ |
| Fontanelle, binary | ✓ | | |
| **B) 6-60-month models** | **M6PD-C$_{6-60}$** | **M6PD-CS$_{6-60}$** | **M6PD-A$_{6-60}$** |
| **Average Cross-Validation Performance** | | | |
| Specificity | 0.57 | 0.59 | 0.54 |
| AUROC | 0.73 | 0.74 | 0.75 |
| PPV | 0.09 | 0.09 | 0.08 |
| NPV | 0.98 | 0.98 | 0.98 |
| PRAUC | 0.16 | 0.15 | 0.15 |
| Brier Score | 0.04 | 0.04 | 0.04 |
| **Full Dataset Performance** | | | |
| Specificity | 0.53 | 0.58 | 0.55 |
| AUROC | 0.75 | 0.76 | 0.77 |
| PPV | 0.08 | 0.09 | 0.08 |
| NPV | 0.98 | 0.98 | 0.98 |
| PRAUC | 0.17 | 0.16 | 0.17 |
| Brier Score | 0.04 | 0.04 | 0.04 |
| **Variables** | | | |
| Age, months | ✓ | ✓ | ✓ |
| Haemoglobin, g/dl | | | ✓ |
| HIV, binary | | ✓ | ✓ |
| How long since last admission, categorical | ✓ | ✓ | ✓ |
| MUAC, mm | ✓ | ✓ | ✓ |
| SpO$_2$, % | ✓ | ✓ | ✓ |

*(Continued)*

**Table 2.** (*Continued*)

| A) 0-6-month models | M6PD-C$_{0-6}$ | M6PD-CS$_{0-6}$ * | M6PD-A$_{0-6}$ * |
|---|---|---|---|
| Water source, categorical | | ✓ | ✓ |
| Weight for age z-score | ✓ | ✓ | ✓ |
| Abnormal BCS, binary | ✓ | | |
| Respiratory rate, bpm | ✓ | | |
| Temperature, ˚C | ✓ | | |
| Boil/disinfect/filter water | | ✓ | |

* Note, the 0-6-month final reduced (M6PD-A$_{0-6}$) and final clinical and social model (M6PD-CS$_{0-6}$) are identical since the same variables were selected.

Abbreviations: AUROC = area under the receiver operating curve; BCS = Blantyre coma scale; bpm = breaths per minute; HIV, human immunodeficiency virus; MUAC = mid-upper arm circumference; NPV = negative predictive value; PPV = positive predictive value; PRAUC = area under the precision-recall curve; SpO$_2$ = oxygen saturation.

disinfection practices replaced by hemoglobin; performance metrics were nearly identical (**Table 2B** and **S8 Text**).

## Discussion

Using four large, objective-driven, prospective cohorts of under-5 children admitted with suspected sepsis, we derived and internally-validated prediction models for post-discharge mortality using only admission data. Their performance to predict mortality up to six months post-discharge was good, suggesting potential utility to improve post-discharge outcomes by linking individual risk to interventional intensity [25]. Data-driven, child-centred approaches to post-discharge care have been strongly advocated for [5,26,27]. Our robust, cross-validated models utilized data from multiple sites, captured over eight years, and should spur focus on external validation outside Uganda.

Several recent studies have shown that post-discharge mortality can be closely linked to a variety of key risk factors, such as malnutrition and disease severity [4–6]. Our results affirm this through formal model development using varied sets of few, objective, and easy-to-collect variables typically available in most settings where such models would be used. In a model deployment context, however, the general approach of developing a single model may not always be sufficient since missingness at the point-of-care may be common. Having multiple simplified models with similar performance, as we saw in our models, may help alleviate these kinds of logistical barriers to implementation [28,29].

Without an effective intervention, risk prediction has limited utility. Understanding discharge as a dynamic process encompassing the time between admission and re-integration into community care is integral to our focus on admission factors [30]. Early identification allows post-discharge risk to be incorporated into discharge planning from the outset. Significant challenges in preparing caregivers for discharge and the transition home have been identified, suggesting that early planning is an essential component of effective peri-discharge care [30].

Choosing risk probability thresholds to classify post-discharge mortality as a binary outcome depends on many factors, including availability of human resources, baseline risk, risk tolerance, and impact on patients/caregivers. Though the thresholds chosen may prove useful in some settings, choice of both the threshold and number of thresholds must be informed by local context and constitutes a critically important consideration for deployment of this, or any, risk model [31].

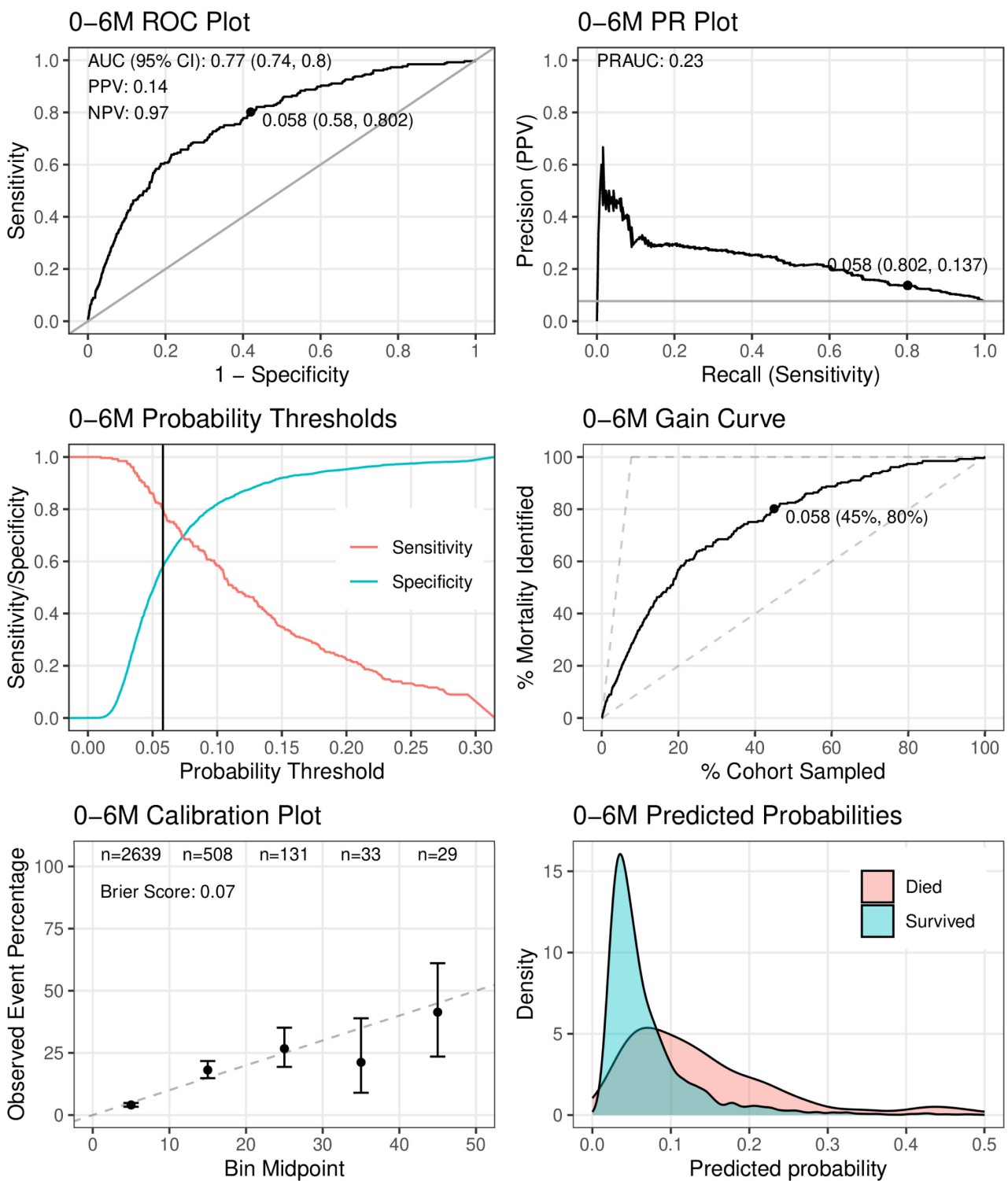

**Fig 3. Performance of the final clinical model for 0–6 months (M6PD-C$_{0-6}$) on the full dataset.** The points on the receiver operating characteristic (ROC), precision recall (PR), and gain curve plots indicate co-ordinates for the probability threshold at sensitivity = 80%, with positive predictive value (PPV) and negative predictive value (NPV) also reported at this threshold.

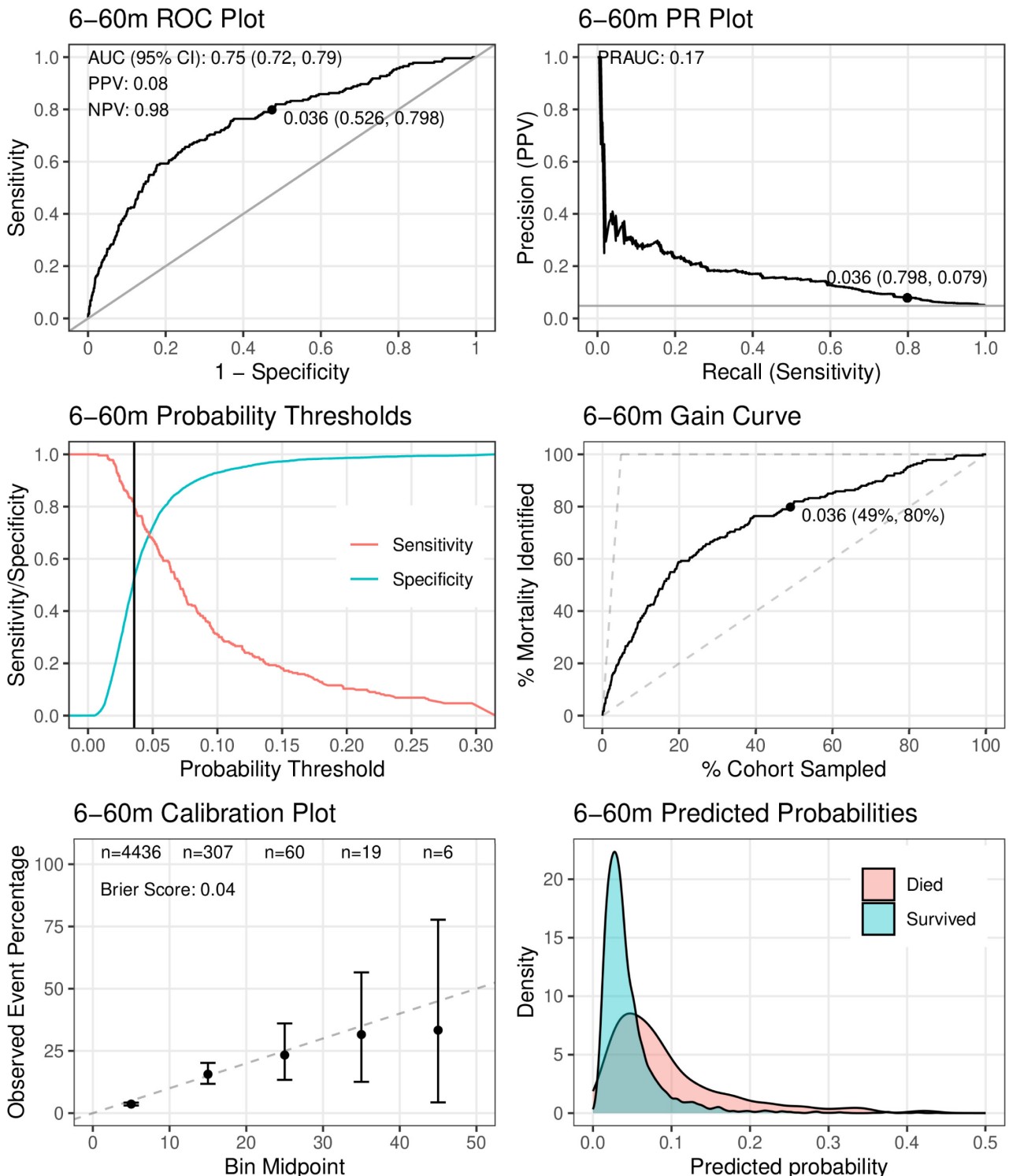

**Fig 4. Performance of the final clinical model for 6–60 months (M6PD-C$_{6-60}$) on the full dataset.** The points on the receiver operating characteristic (ROC), precision recall (PR), and gain curve plots indicate co-ordinates for the probability threshold at sensitivity = 80%, with positive predictive value (PPV) and negative predictive value (NPV) also reported at this threshold.

Although internal validation can justify using models within the region in which they are derived, external validation using different data sources (ideally several) from different regions is essential [32], using both existing and future data [5]. Consequently, we have several prospective studies underway, and will establish data sharing agreements with other collaborators to enable use of their collected data. However, not every conceivable implementation region for any given model can be subjected to external validation. A more pragmatic approach is developing a region-specific model-updating process, integrated over the life-course of the model. Calibration drift due to secular trends, the measured impact of the model itself, and peculiarities of each individual site are key considerations in model deployment [33]. Digitization of the healthcare system will help establishing these processes [34].

As health systems in low-income countries increasingly adopt electronic health records, incorporating algorithms to augment care decisions has tremendous potential to improve outcomes and facilitate adoption of these digital systems [35,36]. Using routinely-collected variables can allow models to run without additional user input and automatically prompt follow-up guidance to the medical team and patient, encouraging adoption and linkage to interventional programs. Furthermore, such systems can report baseline risk data and, when linked to follow-up programs, data on readmission and mortality to national-level health management information systems, such as DHIS2 [37]. These data can be used in model calibration and updating, ensuring site-specific validity. Contextually-validated digital clinical decision support systems utilizing risk algorithms are increasingly recognized as essential to achieving universal health coverage, especially in low- and middle-income countries [38,39].

## Limitations

This study has several limitations. While our models performed well with internal cross-validation, demonstrating good performance in planned external validation is essential to encourage adoption. Second, our models do not accommodate missing data for predictor variables. While missing data rates were very low, this is unlikely to represent true rates of missingness in real-world practice. We developed a family of models, varying in number and type of predictors, which produced similar performance, to partially address this limitation. Future research will explore more robust methods for addressing missing data, including building sub-models to allow for every possible combination of missing variable [29]. Third, these models were developed in the absence of a proven program to utilize a risk-based approach to care, limiting their current utility. While merely knowledge of individual risk can change behaviour and may influence provision of peri-discharge care, risk-informed approaches to follow-up care are also currently under investigation [40]. Fourth, calibration was good at most observed risk levels, but there were very few patients with predicted risk greater than 40–50%, so calibration beyond these probabilities could not be assessed. Regardless, our models should perform adequately for implementation purposes using the optimal threshold cut-offs identified. Finally, the added value of these models may be questioned in the light of previously published models [40–42]. Our models were based on purposively built cohorts, with *a priori* stakeholder engagement regarding relevant variables and their measurement timing, and were uniquely developed within the clinical rubric of suspected sepsis, which is increasingly recognized as a global health priority.

## Conclusion

Post-discharge mortality in the context of suspected sepsis occurs frequently in children under five years old, but those at highest risk can be identified using simple clinical criteria, measured at admission. Being able to select from a range of prediction models, with similar performance

parameters, may support wider implementation of digital risk-stratification tools in different clinical settings. Future work must focus on both external validation as well evaluation of how risk-stratified care can improve post-discharge outcomes.

## Supporting information

**S1 Checklist.**
(DOCX)

**S1 Text. Details of study cohorts and variables used for the full and intermediary models.**
(DOCX)

**S2 Text. Statistical methods and analyses.**
(DOCX)

**S3 Text. Intermediary clinical variable models–variable importance.**
(DOCX)

**S4 Text. Intermediary clinical and social variable models–variable importance.**
(DOCX)

**S5 Text. Intermediary any variable models–performance metrics, coefficients and variable importance.**
(DOCX)

**S6 Text. Final clinical variable models, M6PD-C$_{0-6}$ and M6PD-C$_{6-60}$ –performance metrics and coefficients.**
(DOCX)

**S7 Text. Final clinical and social variable models, M6PD-CS$_{0-6}$ and M6PD-CS$_{6-60}$ –performance metrics and coefficients.**
(DOCX)

**S8 Text. Final any variable models, M6PD-A$_{0-6}$ and M6PD-A$_{6-60}$ –performance metrics and coefficients.**
(DOCX)

**S9 Text. Literature review.**
(DOCX)

## Acknowledgments

We would like to acknowledge all past and present members of the Smart Discharges Research program for their efforts in data collection, administration, logistics support, and all study activities, including but not limited to: Tumwebaze Godfrey, Agaba Collins, Tumukunde Goreth, Naturinda Mackline, Assimwe Abibu, Nakafero Joan, Kiiza Israel, Kitenda Julius, Kamba Ayub, Kuguminkiriza Brenda, Kabajasi Olive, Kembabazi Brenda, Happy Annet, Tusingwire Fredson, Nuwasasira Agaston, Ankatse Christine, Naturinda Rabecca, Nabawanuka Abbey Onyachi, Kamazima Justine, Kairangwa Racheal, Ounyesiga Thomas, Mwoya Yuma, Twebaze Florence, Bulage Mary, Tugumenawe Darius, Tuhame Dyonisius, Twesigye Leonidas, Kamusiime Olivia, Ainembabazi Harriet, Abaho Samuel, Nakabiri Zaituni, Naigaga Shaminah, Kisame Zorah, Babirye Clare, Kayegi Maliza, Opuko Wilson, Mwaka Savio, Baryahirwa Hassan, Mutungi Alexander, Charlene Kanyali, Catherine Kiggundu, Alexia Krepiakevich, Brooklyn Nemetchek, Jessica Trawin, Maryum Chaudhry, Peter Lewis, Rishika Bose, Sahar Zandi

Nia, Tamara Dudley, and Cherri Zhang. Without their effort and support, this study would not have been possible.

## Author Contributions

**Conceptualization:** Matthew O. Wiens.

**Data curation:** Matthew O. Wiens, Dustin Dunsmuir, Martina Knappett.

**Formal analysis:** Matthew O. Wiens, Vuong Nguyen, Jeffrey N. Bone, Clare Komugisha, Mellon Tayebwa, Douglas Mwesigwa, Martina Knappett.

**Funding acquisition:** Matthew O. Wiens, Jeffrey N. Bone, Elias Kumbakumba, Abner Tagoola, Celestine Barigye, Jesca Nsungwa, Charles Olaro, J. Mark Ansermino, Niranjan Kissoon, Joel Singer, Charles P. Larson, Pascal M. Lavoie, Peter P. Moschovis, Stefanie Novakowski, Nathan Kenya Mugisha, Jerome Kabakyenga.

**Methodology:** Matthew O. Wiens, Elias Kumbakumba, Abner Tagoola, Celestine Barigye, Charles Olaro, J. Mark Ansermino, Niranjan Kissoon, Joel Singer, Charles P. Larson, Pascal M. Lavoie, Dustin Dunsmuir, Peter P. Moschovis, Nathan Kenya Mugisha, Jerome Kabakyenga.

**Project administration:** Matthew O. Wiens, Elias Kumbakumba, Clare Komugisha, Mellon Tayebwa, Douglas Mwesigwa, Jerome Kabakyenga.

**Resources:** Stephen Businge.

**Supervision:** Stephen Businge, Abner Tagoola, Sheila Oyella Sherine, Emmanuel Byaruhanga, Edward Ssemwanga, Celestine Barigye, Dustin Dunsmuir, Clare Komugisha, Mellon Tayebwa, Douglas Mwesigwa.

**Visualization:** Vuong Nguyen, Nicholas West.

**Writing – original draft:** Matthew O. Wiens, Vuong Nguyen, Nicholas West.

**Writing – review & editing:** Matthew O. Wiens, Vuong Nguyen, Jeffrey N. Bone, Elias Kumbakumba, Stephen Businge, Abner Tagoola, Sheila Oyella Sherine, Emmanuel Byaruhanga, Edward Ssemwanga, Celestine Barigye, Jesca Nsungwa, Charles Olaro, J. Mark Ansermino, Niranjan Kissoon, Joel Singer, Charles P. Larson, Pascal M. Lavoie, Dustin Dunsmuir, Peter P. Moschovis, Stefanie Novakowski, Clare Komugisha, Mellon Tayebwa, Douglas Mwesigwa, Martina Knappett, Nicholas West, Nathan Kenya Mugisha, Jerome Kabakyenga.

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
